Ribovirus classification by a polymerase barcode sequence

Babaian Artem 1 2
http://orcid.org/0000-0001-7355-2541 Edgar Robert 3 robert@drive5.com
1 St Edmunds College , Cambridge , United Kingdom
2 Department of Haematology, University of Cambridge , Cambridge , United Kingdom
3 Corte Madera, California , United States of America
Grande-Pérez Ana
Electronic publication date: 2022 Oct 13
Publication date: 2022
Volume: 10
Electronic Location ID: e14055
Received 2022 Apr 26; Accepted 2022 Aug 24
Copyright: © 2022 Babaian and Edgar
Copyright year: 2022
Copyright holder: Babaian and Edgar
License: This is an open access article distributed under the terms of the Creative Commons Attribution License, which permits unrestricted use, distribution, reproduction and adaptation in any medium and for any purpose provided that it is properly attributed. For attribution, the original author(s), title, publication source (PeerJ) and either DOI or URL of the article must be cited.
License URL: https://creativecommons.org/licenses/by/4.0/

Erratum in: Correction: Ribovirus classification by a polymerase barcode sequence 10 9 8 2024 e14055/correction-1 PeerJ 10.7717/peerj.14055/correction-1 PMC11321492 39139825
Keywords: RNA virus, RNA-dependent RNA polymerase, Virus classification, Virus evolution

Funding: Canadian Institutes of Health Research (CIHR) Banting Postdoctoral Fellowship Artem Babaian was supported by a Canadian Institutes of Health Research (CIHR) Banting Postdoctoral Fellowship. The funders had no role in study design, data collection and analysis, decision to publish, or preparation of the manuscript.

==============================
RNA viruses encoding a polymerase gene (riboviruses) dominate the known eukaryotic virome. High-throughput sequencing is revealing a wealth of new riboviruses known only from sequence, precluding classification by traditional taxonomic methods. Sequence classification is often based on polymerase sequences, but standardised methods to support this approach are currently lacking. To address this need, we describe the polymerase palmprint, a segment of the palm sub-domain robustly delineated by well-conserved catalytic motifs. We present an algorithm, Palmscan, which identifies palmprints in nucleotide and amino acid sequences; PALMdb, a collection of palmprints derived from public sequence databases; and palmID, a public website implementing palmprint identification, search, and annotation. Together, these methods demonstrate a proof-of-concept workflow for high-throughput characterisation of RNA viruses, paving the path for the continued rapid growth in RNA virus discovery anticipated in the coming decade.

Introduction

RNA viruses encoding a polymerase gene (riboviruses) dominate the known eukaryotic virome. High-throughput sequencing is revealing a wealth of new riboviruses known only from sequence evidence (Edgar et al., 2022), precluding classification by traditional taxonomic methods. Sequence classification is often based on polymerases (Koonin et al., 2020), but standardised methods to support this approach are currently lacking. To address this need, we describe the polymerase palmprint, a segment of the palm sub-domain robustly delineated by well-conserved catalytic motifs. We present an algorithm, Palmscan, which identifies palmprints in nucleotide and amino acid sequences; PALMdb, a collection of palmprints derived from public sequence databases; and palmID, a public website implementing palmprint identification, search, and annotation. Together, these methods demonstrate a proof-of-concept workflow for high-throughput characterisation of RNA viruses, paving the path for the continued rapid growth in RNA virus discovery anticipated in the coming decade.

Microbe classification in the metagenomics era

The deluge of new data from metagenomic sequencing has prompted the development of new classification approaches based on so-called marker, tag, or barcode regions of microbial genomes such as the internal transcribed spacer region for fungi (Abarenkov et al., 2010) and 16S ribosomal RNA gene for the domains Bacteria and Archaea (Pruesse et al., 2007). In virology, classification on the basis of marker genes, typically polymerases, is well established (Koonin et al., 2020; Obbard et al., 2020; Shi et al., 2016; Zayed et al., 2022; Starr et al., 2019), but progress is impeded by a lack of standardised sequence analysis methods and databases. Here, we propose a standard barcode for viral polymerases, focusing primarily in this work on RNA dependent RNA polymerase (RdRP) the hallmark for Orthornavirae.

Viral polymerase

RdRP and reverse transcriptase (RT) belong to the template-dependent nucleic acid polymerase superfamily; the RdRP structure resembles a grasping right hand with thumb contacting finger (Mönttinen et al., 2014). Interior-hand surface regions which are involved in nucleotide selection or catalysis are strongly conserved, in particular short motifs conventionally designated by letters A through G, although not all of these motifs are ubiquitous among polymerases (reviewed in Jia & Gong (2019)). Motifs A, B and C are in the palm sub-domain and are well conserved in most known RdRPs and RTs (te Velthuis, 2014). A and C contain essential aspartic acid residues which coordinate the Mg+/Mn+ cation for catalysing phosphodiester bond formation, while B contains an almost perfectly conserved glycine required for nucleotide selection. The motifs appear in ABC (canonical) order in the primary sequence of most known polymerases, but the active site sequence is permuted into CAB order in several independent lineages (Gorbalenya et al., 2002; Sabanadzovic, Abou Ghanem-Sabanadzovic & Gorbalenya, 2009) (Figs. 1 and 2).

Figure 1 Conservation of catalytic motifs in three divergent RdRp structures in the Protein Data Bank.

Coronaviridae (COV, virus: SARS-CoV-2, pdb: 7CYQ.) is from a positive-strand RNA virus. Reoviridae (REO, virus: Mammalian orthoreovirus 3 Dearing, pdb: 1N1H) is from adouble-stranded RNA virus, and Permutotetraviridae (PER, virus: Thosea asigna virus, pdb: 5CYR) is apositive-stranded RNA virus with permutation of motif C. RdRp domains defined by PFAM (COV: RdRP_1, TAV: DNA/RNA pol_sf, and REO: RdRP_5) are shown within the open reading frame for REO and PER, and the mature polyprotein cleavage peptide from the 7,096aa ORF1ab for COV.

Figure 2 (A) The palmprint segment is a ~100aa region in the active site of the polymerase domain.

Motifs A, B and C are well-conserved; the intervening V1 and V2 regions are more variable. (B) Sequence logos for the five established RdRp-containing ribovirus phyla (Duplorna=Duplornaviricota, etc.) and one for RTs in Artverviricota; a PSSM is constructed corresponding to each logo. (C) Palmscan alignment for NC_009224.1 palmprint showing motifs in canonical ABC order. (D) Alignment for NC_040813.1 withpermuted motifs in CAB order.

Amino acid identity can be as low as 10% between diverged species (Bruenn, 2003). The boundary of the polymerase domain is often unclear as the gene may be embedded in a longer open reading frame (ORF) together with other functional domains. Significant sequence similarity with a known viral polymerase is not sufficient to establish viral origin or polymerase function as viral RdRPs can be integrated into host germline DNA via reverse transcription, becoming an endogenous viral element (EVE) (Holmes, 2011; Feschotte & Gilbert, 2012). Viral RTs are ubiquitously inserted into host germlines, losing function over time and leaving “fossils” with recognisable sequence similarity (Bock & Stoye, 2000).

Palmprint definition

Ideally, a barcode should be delineated by conserved motifs, and the segment defined by these motifs should be globally homologous across all, or a large majority, of known sequences. These requirements enable automated identification of the barcode segment, commensurate estimates of sequence divergence from pair-wise global alignments, and phylogenetic tree estimation from global multiple alignments. Global alignments enabled by consistently trimmed segments are more robust and consistent under variation in algorithms and parameters such as gap penalties compared to local alignment methods such as BLAST (Altschul et al., 1997). Approximate global homology across the barcode ensures that sequence length variation is due to indel events and that the intervening sequence is under comparable constraints in different clades, thus providing a uniform standard for measuring evolutionary distance. With these goals in mind, we defined the polymerase palmprint to be the minimal segment containing all letters of the A, B and C motifs (Figs. 1–3). In domains where the motifs appear in canonical order, the palmprint extends from the first letter of A to the last letter of C; in CAB permuted domains it extends from the first letter of C to the last letter of B. We excluded motifs D through G for delineating a barcode because they are less well conserved and are sometimes unidentifiable or absent.

Figure 3 Defining RdRp boundaries for sequence-based classification.

Schematic depiction of methods for defining RdRp segment boundaries for sequence analysis. As shown at the top, RdRp may be embedded in a multi-gene ORF (see also Fig. 1). Below are three alternative RdRp boundary schemes defined by Wolf et al. (2018) (“Wolf2018”), Zayed et al. (2022) (“Zayed2022”), and Edgar et al. (2022) (“Edgar2022”), respectively. Wolf2018 attempted to identify approximately full-length genes, discarding fragments unless they are close to full-length. This scheme is problematic because RdRp is often found in a longer ORF with other functional domains, and in such cases the boundary of the RdRp is often unclear. Zayed2022 used a similar scheme while additionally allowing fragments. Allowing fragments allows more sequences to be included but is problematic for classification because pairs with little or no overlap may be assigned to different vOTUs even if they belong to the same species. Edgar2022 used palmprints, a short segment of RdRp with well-defined boundaries.

Palmprint identification

Given a structure, the palm domain can be visually identified by its grasping right-hand shape, and the A, B and C motifs can be visually identified by their secondary structure and position within the palm (Fig. 1). Thus, the palm domain and its palmprint appear amenable to computational identification from structure, but we do not explore this direction further here. The sequences of A, B and C are well-conserved within major groups (te Velthuis, 2014), each comprising around 12 to 15aa where indels are highly suppressed and are conveniently represented as sequence logos (Fig. 2). Such logos facilitate visual identification of motifs in an amino acid sequence.

palmID: palmprint analysis suite

The known virome is growing and modern computational virology infrastructure should anticipate the integration of viral sequences, and their meta-data, numbering in the billions of records by the end of the decade. To demonstrate the functional application of palmprints in database integration, we created palmID (https://serratus.io/palmid), a free web-analysis tool (also available as a downloadable container) which receives a known or novel RdRP sequence as input and aggregates sequence and meta-data from similar viral RdRPs (Fig. 4).

Figure 4 Overview of palmID and procedurally generated figures (interactive version: https://serratus.io/palmid?hash=ruby).

(A) Workflow, (B) quality control, (C) geospatial map, and (D) matching palmprints in PALMdb.

Methods

Palmscan algorithm

Sequence logos can be formalised as position-specific scoring matrices (PSSMs) which enable efficient computational search for ungapped matches (Stormo et al., 1982). We implemented this approach in a new algorithm, Palmscan, which automates identification of palmprints. Briefly, Palmscan aligns motif PSSMs for known groups to a nucleotide or amino acid sequence and reports hits where A, B and C motifs have high log-odds scores and are separated by distances comparable to known palm domains. Matching each motif separately enables identification of permuted domains. See Figs. 1C and 1D for example Palmscan alignments.

Position specific scoring matrices

We created PSSMs (Stormo et al., 1982) to recognise A, B and C motifs. A PSSM P[i,j],i=1…L,j=1…20 represents a multiple alignment of length L columns by a log-odds score for each possible amino acid in every column. Letters found with high frequency at a given position are assigned positive scores, while unobserved and low-frequency letters have negative scores. The score of Pi aligned to the kth letter in a query sequence Q is P[i,Qk], and the total score of the alignment is the sum of scores

(1) ∑i=1LP[i,Qs+i],

where s is the starting position in Q. Gaps are not permitted in the alignment. Log-odds scores were calculated from multiple sequence alignments (MSAs) for each motif assuming background amino acid frequencies from BLOSUM62 (Henikoff & Henikoff, 1992) with pseudo-count frequency 0.1. Seven sets of PSSMs were constructed: one for each recognized phylum where RdRP is commonly found, i.e., Duplornaviricota, Kitrinoviricota, Lenaviricota, Negarnaviricota and Pisuviricota; one for reverse transcriptases in Artverviricota; and one for permuted members of the Birnaviridae family, which have distinctly different motifs. MSAs for each motif in each set were constructed as follows. First, a small seed alignment was constructed by hand. The resulting PSSM was then used to search polymerase-containing datasets. This process was iterated until no further improvement was obtained. This may have resulted in some degree of over-training to known polymerases, but sensitivity was not our primary goal; we preferred to ensure that positive PSSM alignment scores were strongly predictive of a valid viral motif.

Segment length distributions

In a palmprint with canonical ABC motif order, we defined the V1 segment to be the sequence between A and B; similarly V2 is the sequence between B and C. In a permuted palmprint with CAB order, V1 is the sequence between C and A; V2 is the sequence between A and B. The V1 and V2 (variable) segments are less well conserved than the motifs. The lengths of the V1 and V2 segments and the total length of the palmprint vary within constraints imposed by structure and function; length distributions were measured for Orthornavirae RefSeqs (Fig. 5).

Figure 5 Lengths of the V1 and V2 variable regions and palmprint segment.

Distributions were measured on full-length RefSeq Orthornavirae genomes.

Palmprint score

The first step of Palmscan is to align all PSSMs to all positions in the query sequence, which is six-frame translated in the case of a nucleotide query. The highest-scoring alignment for each PSSM is recorded, and the motif set with highest total log-odds score is identified. If the order is not ABC or CAB, the query is rejected. Otherwise, the lengths of V1, V2 and putative palmprint are determined. A palmprint score is derived, starting with the sum of log-odds scores of the three PSSMs in the highest-scoring set, adjusted by a series of heuristics which are designed to quantify qualitative judgements that might be made by a human expert. For example, if any PSSM log-odds score is <2 then the alignment is rejected; if the length of V1 is less than 35aa then a penalty of 5 is subtracted from the score. These heuristics were developed through an iterative process of manual review and improvement of Palmscan predictions on ribovirus RefSeqs and decoys. If the final score is ≥20, the palmprint is reported as a high-confidence hit.

Species identity threshold

We sought a palmprint identity threshold such that two viruses with palmprints having higher (lower) identity tend to belong to the same (different) species. We tuned the threshold by clustering all palmprints of ICTV species into species-like Operational Taxonomic Units (sOTUs, (Urayama et al., 2018; Gustavsen et al., 2014; Edgar et al., 2022)) using UCLUST (Edgar, 2010) at a range of thresholds, selecting 90% identity as the threshold which balanced “lumping” and “splitting” of species (Fig. 6).

Figure 6 Identity threshold tuning.

(A) Number of clusters obtained by clustering RdRP palmprints of 2,048 recognised ICTV species at identity thresholds 97%, 96% … 85%. (B) Number of species that are split over multiple OTUs, lumped together with one or more other species into a single OTU, both lumped and split (Lmp+Spl), or pure (not lumped or split). The best fit of number of clusters to number of species is obtained at 90% identity.

Prediction accuracy metrics

We use positive predictive value (PPV) and false discovery rate (FDR) as our accuracy metrics for palmprint identification and classification algorithms. PPV is the number of correct predictions divided by the total number of predictions; FDR is the number of incorrect predictions divided by the total number of predictions. By design, queries which are unclassified by a prediction method are not considered by these metrics.

Reference palmprints

We used the following datasets in validation and training. Wolf18 is the RNAvirome.S2.afa file from Wolf et al. (2018). PF00680_RdRP_1, PF00978_RdRP_2, PF00998_RdRP_3 and PF02123_RdRP_4 are full alignments of PFAM (Bateman et al., 2004) RdRP models; similarly PF00078_RT1 and PF07727_RT2 are PFAM RT alignments. Genomes is the set of RefSeq complete genomes from ribovirus phyla that typically contain RdRP (see Introduction). GB241nt is all nucleotide sequences from GenBank (Benson et al., 2012) v241, GB241aa is all translated CDSs from GenBank v241, NR is the NCBI non-redundant protein database (Pruitt, Tatusova & Maglott, 2005). Palmscan hits from all these sources were combined into a set of reference sequences, excluding those matching RTs. Those annotated by the source database as non-viral were retained to support discrimination of viral RdRP palmprints from RTs and non-viral palmprint-like sequences.

Decoy set of non-RdRP sequences

We created a consolidated set of non-RdRP sequences (Decoy) to aid the development and validation of Palmscan, including the following amino acid sequences clustered at 97% identity: UniProt proteomes for human (UP000005640), yeast (UP000002311), E. coli (UP000000558); all retroviral and DNA-viral sequences from GenBank; PF00078_RT1 and PF07727_RT2; the training set of validated RT developed for myRT (Sharifi & Ye, 2021), and disordered proteins from https://disprot.org (Hatos et al., 2020). Manual inspection of the following 10 sequences identified as high-confidence RdRP by Palmscan were confirmed by InterPro (Hunter et al., 2009) and BLAST (Altschul et al., 1997) to be RdRP and discarded: CZQ50745.1, AFH02745.1, AFH02746.1, ADO67072.1, pdb|4TN2|A, YP_009506261.1, ADE61677.1, YP_009551602.1, AXY66749.1, and AWS06671.1.

Results

Palmscan accuracy

Results on validation datasets are shown in Table 1. The sensitivity of Palmscan, measured as the fraction of positive sequences with a hit, is 87%. Combined, the results of the positive and negative datasets imply that on this data, the positive predictive value of Palmscan is 0.9989 and the false discovery rate is 0.0010. Palmscan reported 64 false positive palmprints on a random nucleotide sequence of length 1011 letters (100G).

Table 1 Palmscan accuracy.

Set	Pos/Neg	N	N_ps	N_dmd	
PF00680_RdRP_1	Pos	795	760	795	
PF00978_RdRP_2	Pos	397	379	396	
PF00998_RdRP_3	Pos	205	194	204	
PF02123_RdRP_4	Pos	216	191	214	
Genomes	Pos	1,959	1,592	1,609	
PF00078_RT1	Neg	46,876	0	13,102	
PF07727_RT2	Neg	12,037	0	1	
Decoy	Neg	296,536	4	74,969	
Note:

N is the number of sequences in a dataset, Pos/Neg indicates whether the dataset is RdRP-positive (most or all sequences should contain RdRPs), or RdRP-negative (most sequences should not contain RdRP). N_ps is the number of palmprints reported, and N_dmd is the number of diamond hits with E-value < 10−5. Notice the large numbers of diamond hits to non-RdRP sequences; in particular RTs. On the positive sets, Palmscan reports hits on a total of 3,116/3,572 = 87% of sequences. See methods for description of datasets.

Score threshold

We set the minimum score to report a high-confidence RdRP palmprint to 20 after review of the score distributions shown in Fig. 7.

Figure 7 High-confidence palmprint score threshold.

Distribution of RdRP palmprint scores on non-RdRP decoy set (top) and full PFAM RdRP alignments (bottom). The score threshold was set to 20 to discriminate RdRP polymerases with high confidence.

Identity threshold

We set the clustering threshold to 90% after review of the results shown in Fig. 6, which shows that the number of clusters is close to the number of species at this identity.

PALMdb

We deployed Palmscan to identify palmprints in public databases, creating a new database PALMdb as a resource for virus classification (https://github.com/rcedgar/palmdb) and web interface palmID (https://serratus.io/palmid) for search and classification of nucleotide and amino acid sequences containing RdRP.

Example palmID analysis of Rubiviruses

Figure 4 shows an example palmID analysis of a Rubivirus palmprint starting from the non-structural polyprotein as input (accession: NP_062883.2), the full-report can be viewed at https://serratus.io/palmid?hash=ruby. An overview of the palmID workflow where a user-input RdRP sequences are analysed with Palmscan against PALMdb. Each input-palmDB palmprint alignment is weighted by amino-acid identity. Sequencing libraries within the Sequence Read Archive (SRA) (Edgar et al., 2022) are indexed against PALMdb and are retrieved for meta-data aggregation (Fig. 4A). The 99-aa palmprint sub-sequence is extracted and quality control assessment of the input-palmprint: displays palmprint-subsequence coordinates within input-sequence, and input score and component lengths compared against a reference set of ∼15,000 GenBank RdRP sequences (PALMdb v2021-03-02) (Fig. 4B). Matching palmprints are identified in 261 sequencing runs in the SRA from which meta-data are aggregated, this includes geospatial coordinates for 201/261 (77%) samples which can be interactively clicked on and sent for BLAST search, and a release-date timeline (Fig. 4C). An interactive input-palmprint identity and e-value graph reports the 66 matching palmprints from PALMdb (average aa-identity 37.9%, range 28.7–99%) and MUSCLE multiple sequence alignment (not shown). Hits are coarsely striated into species-like to phylum-like categories (Fig. 4D). In addition, a word-cloud of the ‘organism’ data field from the sequencing runs, with size-colour scaled by the input-PALMdb palmprint alignment weighting (percent identity) and organism-level k-mer classification taken from STAT (Katz et al., 2021) (not shown). High-identity Rubella virus palmprints are seen in libraries annotated as ‘viral metagenome’, ‘human skin metagenome’, ‘Homo sapiens’, while more distant palmprints are seen in libraries annotated as ‘bat metagenome’, ‘Amolops mantzorum’ and ‘Plethodon cinereus’.

The user-input palmprint is then aligned against all PALMdb sOTU centroids with DIAMOND (–ultra-sensitive −e 0.00001) to retrieve the set of matching palmprints (up to 500 hits), with matching palmprints coarsely grouped into species-like (>90% identity), genus-like (70–90%), family-like (45–70%), or else phylum-like matches (Fig. 4D). Using the Serratus API, each palmprint match is queried against RdRPs identified from the SRA to retrieve the set of sequencing runs containing input-palmprint or neighbours. Corresponding meta-data are then aggregated. For example, geospatial distribution, sample annotation and virus-associated organisms via k-mer classification are displayed (Figs. 4E and 4F). Interactive figures of this data are generated, and raw data-tables are available for download to the user with a typical running time of around 2 min. Source code for palmID is freely available at https://github.com/ababaian/palmid.

Human Rubella virus (Rubivirus rubellae) is a highly contagious human pathogen which causes “German measles” and has been implicated in congenital birth defects following maternal infection (Cooper, 1985). Two non-human Rubiviruses were recently described: Ruhugu (Rubivirus strelense) infecting cyclops leaf-nosed bats sampled in Uganda, and Rustrela (Rubivirus strelense) infecting yellow-necked field mice in Germany. Rustrela virus was identified after causing acute encephalitis in a donkey, demonstrating that rubiviruses present a risk of zoonotic spillover (Bennett et al., 2020). To exemplify the utility of palmprints in navigating RNA viruses, we sought to identify new Rubiviruses using our palmID web interface. palmID identifies palmprints with Palmscan, searches PALMdb and aggregates virus-associated meta-data from the Sequence Read Archive into a user-facing report (Fig. 8). The result of a two-minute search (results: https://serratus.io/palmid?hash=ruby) reported five genus-like virus sequence matches to Rubella: Ruhugu (86.9% palmprint identity to Rubella), Rustrela (76.8%) and three uncharacterised viruses. We refer to the novel viruses as Ruche Rubivirus (83.8%), Rumple Rubivirus (75.8%), and Ruffle Rubivirus (72.7%), respectively (Table S1) and assembled the libraries reported to contain these palmprints (Fig. 8). Ruche virus was identified from an RdRP fragment in a sample annotated as originating from a Greater Horseshoe Bat (Rhinolophus ferrumequinum) collected in 2013 in Shanxi, China (Wu et al., 2016). Rumple virus was identified in four samples originating from a North American Red-backed Salamander (Plethodon cinereus) experimentally housed and infected with chytrids (Ellison et al., 2020), and Ruffle virus was observed in Kangting Sucker frog (Amolops mantzorum) sampled no later than 2017 in Sichuan, China (Xia et al., 2018). While these assemblies expand the known diversity of Rubi-like viruses, the primary significance of the latter results is a demonstration that palmprint-based classification of RNA viruses enables rapid search and aggregation of virus-associated meta-data from large databases including geospatial locations, host-organism associations and virus phylogeny.

Figure 8 Rubi- and rubi-like viruses identified by palmID (A) Genome synteny among of Rubiviruses (RV) and related Matonaviruses (MV) showing significant (E < 10−4) protein domain matches.

(B) Parallel phylogenetic tree created from RNA dependent RNA polymerase (RdRP) or concatenated capsid and E2/E1 glycoproteins, inlay showing unrooted RdRP-tree. (C) Protein sequence alignment of the common RdRP fragment with motif A, B, and C highlighted.

Discussion

Our results show that the palmprint is an effective barcode for classifying and navigating Orthornavirae. Such a system can be extended to other riboviruses, namely Pararnavirae and even to the realms of DNA viruses. Palmprint identification by the Palmscan algorithm performs well on two essential tasks in a high-throughput metagenomic analysis of riboviruses: (1) delineating boundaries for a robust barcode, and (2) discriminating RdRPs, which are very likely to be viral, from RTs which may be degraded host insertions. Palmscan was central to a recent petabase-scale data mining effort which revealed 131,957 novel RNA virus species in public data (Edgar et al., 2022), thereby demonstrating the value of our approach. However, it should be noted that we regard Palmscan as a preliminary proof of concept rather than a definitive solution for identifying palmprints from sequence. Notable limitations include a false negative rate of ∼13% on known RdRPs, and its sensitivity to highly diverged groups (novel phyla) is unknown, but likely to be low. While Palmscan, palmID and palmID may be regarded as preliminary proofs of concept, we believe that the conception of the palmprint itself as an RdRP barcode delineated by motifs A and C is a robust definition offering decisive advantages for future work transcending our early implementations.

Conclusion

We have shown that a segment of the viral polymerase palm domain, the palmprint, is well-suited for use as a barcode for Orthornavirae classification, laying a foundation for characterisation of the forthcoming vast expansion of the known planetary virosphere.

Supplemental Information

Supplemental Information 1 Results of palmID search for Rubella virus RdRP.

PalmID search results include hits to SRA accessions, word cloud with keywords, visualization of palmscan alignment and links to downloadable tables.

Click here for additional data file.

Supplemental Information 2 Results of palmID search for Rubella virus RdRP.

SRA accessions, reference sequences and identities obtained using Rubella palmprint as a query.

Click here for additional data file.

Additional Information and Declarations

Competing Interests

Author Contributions

Data Availability

The authors declare that they have no competing interests.

Artem Babaian conceived and designed the experiments, performed the experiments, analyzed the data, prepared figures and/or tables, authored or reviewed drafts of the article, and approved the final draft.

Robert Edgar conceived and designed the experiments, performed the experiments, analyzed the data, prepared figures and/or tables, authored or reviewed drafts of the article, and approved the final draft.

The following information was supplied regarding data availability:

The raw data is available at GitHub: https://github.com/rcedgar/palmdb and https://github.com/rcedgar/palmscan.

The nucleotide sequence data are available in the Third Party Annotation Section of the DDBJ/ENA/GenBank databases (TPA): BK061338–BK061341.

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
