# Peer review of "Ribovirus classification by a polymerase barcode sequence"

_PeerJ, doi:10.7717/peerj.14055_

## Round 0.1 · original submission · Minor Revisions

Congratulations on an excellent paper. Your manuscript has been reviewed by two experts in the field of viral RNA polymerases and metagenomics. I agree with both reviewers that your work is extremely interesting and should be published. However, there are a number of comments that have been raised that should be addressed. In my opinion, all comments, suggestions and corrections are pertinent and, especially in the case of the second reviewer, amply justified. I therefore invite you to submit a letter of response to the reviewers in which you address each and every issue raised by them, as well as a new version of the manuscript in which all their comments have been adequately addressed.

Reviewer 1 ·

Basic reporting

No comment

Experimental design

No comment

Validity of the findings

No comment

Additional comments

This manuscript from Babaian and Edgar describes the development of an algorithm to the identification of novel riboviruses. To this aim, Palmscan identifies nucleotide and amino acid sequence similarity to conserved motifs A, B and C in RNA-dependent RNA polymerases (RdRp). In addition to the identification of novel RNA viruses, including potential zoonotic threats (e.g. new animal Rubellaviruses), the program can be used to a better classification of RNA viruses, currently using whole-sequence polymerases rather than highly-conserved motifs. The algorithm also distinguishes RdRp sequences from reverse transcriptase genes, which favours the discrimination of true riboviruses from retroviruses and from degraded RT genes with a cellular origin (e.g. endogenous RTs inserted in the host genome). I deem this manuscript may be of much interest and incredibly useful for virologists working in the fields of viral phylogenetics, metagenomics in the discovery of new riboviruses, and RNA virologists in general. The manuscript is very well written and perfectly structured, which makes easier its comprehension. Therefore, I fully support its publication in PeerJ. Below, I include some minor suggestions to improve the manuscript:
1. Please provide with a definition for acronyms when they first appear in the text: e.g. OTU, PFAM, etc
2. Line 42. While thumb contacts the fingers domain in RdRps, my understanding is that this is not the case for RTs (open hand shape). Can you clarify?
3. Line 45. Likewise, While RdRps contain A to G conserved motifs, I am not completely certain that all these motifs are found in RTs. My understanding is that RTs only have A to E motifs, although I am not an expert. Please feel free to
4. Figure 6. This figure contains very poor resolution illustrations which makes difficult its interpretation. I’d suggest the authors either or both enlarging the figure and/or removing some of the panels which are less informative (e.g. panels b and f in my humble opinion).
5. line 197. there is a typo, it should be ‘viral metagenome’

·

Basic reporting

The manuscript of Babaian & Edgar entitled “Ribovirus classification by a polymerase barcode sequence” explained, benchmarked, and validated a new bioinformatic pipeline to classify novel RNA viruses by using a selected segment of the viral RdRp (RNA-dependent RNA polymerase) domain sequence (which is present across all RNA viruses replicating via RdRp) as a barcode.

Briefly, in the approach followed by the authors, this segment of the RdRp domain containing only motifs A, B, and C (“palmprint”) is set as a barcode sequence of 100 amino acids. The applied algorithm (Palmscan) is meant to detect and characterize palmprints out of protein encoded by viral genomes and make sure at the same time that this RdRp palmprint signal is not confounded with homologous proteins (such as reverse transcriptases) and many other non-RdRp sequences derived from multiple datasets, in order to avoid the incorporation of false-positive RdRp hits. The complete catalog of these palmprints detected in public databases was dereplicated following different amino acid identities into “OTUs” sequence clusters equivalent to the taxonomic ranks of species, genera, and families. The taxonomic annotations on those taxonomic clusters are derived from a consensus of alignments of the palmprints to established RNA viruses set as standards.

The manuscript represents the companion paper of a work recently published in Nature (Edgar et al. 2022), which introduced a novel open-source platform called Serratus (https://serratus.io/) that is aimed search viral RdRp signal (and hence RNA viral) throughout all available datasets of the internet (e.g.: SRA) against your query sequences. The goal of the manuscript of Babaian and Edgar is to describe the validated method for the systematic classification of RNA viruses using the palmprint sequence.

Babaian & Edgar have identified a key opportunity using a strong computational effort in an era of metagenome-enabled virus discovery, and during a COVID pandemic that has revived the awareness of the zoonotic disease and the emergence of RNA viruses from a vastly uncharacterised RNA virosphere in a globalized world. Although it will be certainly a controversial work from the point of view of most virologists, the tools offered by Babaian & Edgar, are going to be utilised by virologists mining NGS data on a daily basis. This kind of tools are desperately needed in the field. Hence, as stated by the authors, these tools will accelerate the already ongoing virome-enabled discovery. This research aims to contribute to answering essential questions in the field of virology and viral taxonomy.

The findings and conclusions are well stated and consistently justified. Overall, the manuscript is well written using appropriate technical language and the text is well organized, and it meets the standards of the journal given the case of study and the goals.

Nevertheless, the manuscript shows a few deficiencies that need to be addressed to place this proposed tool into a suitable context, mostly regarding the introduction and discussion, and conclusions sections.

1) Appropriate literature context is lacking in the introduction section with respect to previous environmental RNA virus efforts, which are relevant given that the focus of the authors was the ultra-sensitive discovery of RNA viral sequences. At least certain recent works bringing highly divergent RdRp sequences found in different sources such as invertebrates (Obbard et al. 2020; Shi et al. 2016), oceans(Urayama et al. 2018; Wolf et al. 2020; Zayed et al. 2022), and soils (Starr et al. 2019) must be cited to place the readers into an appropriate context of the state of art. Given the comment on particularly divergent RdRp sequences derived from novel RNA virus phyla, authors must cite Zayed et al. (2022) in L245, which recently challenged the established 5-phyla system proposed initially by (Wolf et al. 2018). The same literature context is lacking for previous efforts to establish viral OTUs (Urayama et al. 2018; Gustavsen et al. 2014; Zayed et al. 2022).
2) It is obvious that the traditional “modus operandi” for viral classification, which was based on phenotypic characterization of viral isolates (virion structure, host range, pathology), is being challenged by the avalanche of novel viral sequences derived from recent sequencing technologies (particularly from environmental sampling). Those large sequence datasets cannot be classified using those conventional criteria, and hence classification based only on sequence comparison represents a tempting, operational answer to the problem. This has been extensively explored and discussed previously (Simmonds 2015; Lauber and Gorbalenya 2012; Wolf et al. 2018). Nevertheless, this solution is well far away from current methods for virus classification, and most virologists are attached to those traditional criteria that are perceived as more consistent than only purely molecular sequences, or even less information if we use only a segment of approximately 100 aa of the RdRp domain sequence. Given this significant departure, one could not be surprised if find resistance to abandoning viral phenotype to classify RNA viruses. Hence, the authors are unfortunately obliged to (at least) state explicitly this discrepancy in the virology community.
3) Related to the previous point (classification only using a segment of the RdRp domain), there are cases where classification derived from sequence similarities and evolutionary histories are incongruent with classifications integrating biological phenotypes of viruses (likely due to recombination and reassortment). Even more importantly for this study, the whole strategy proposed by Babaian & Edgar is assuming that classification thresholds for viral taxa are the same across different evolutionary lineages (as also happened with tools such as PASC and DEmARC). Unfortunately, we know that the reality is more complicated and that different lineages of RNA viruses can show diverse evolutionary rates. Of course, I understand that the effort of the authors was aimed to provide a series of “systematic” thresholds of sequence comparisons to get a reliable taxonomy. Given that this is still a hot topic of discussion among current virologists, and despite the compelling evidence provided by the authors for their pragmatic goal, they are again obliged to clearly specify this concern in the discussion for the sake of transparency.
4) Given the strong computational component of this work, the authors may make a more compelling case if they could provide a demonstration of the detection power of RNA viruses against other methods that were utilised previously in environmental RNA virology. Specifically, I suggest authors compare their detection/classification method with those based on HMM or BLAST search to find viral RdRp sequences. Previous works regarding these search methods are cited in point 1 above.
5) Motifs A, B, and C (that in combination with separating segments constitute the “palmprint”) were selected because of their permanent presence and their high degree of conservation across alignments. Nevertheless, I missed the due value of the rest of the motifs of the RdRp domain sequence, which are also critical for functionality, in the explanation provided. Motifs D and G are also important and very frequently present. It is true that motif E is absent in the bacteriophage Qbeta, but mostly present. The long motif F is not even mentioned in the manuscript. Those motifs are undoubtedly additional sequence information that can be used to identify viral sequences and classify them. The exclusion of the remaining 4-5 motifs of the RdRp domain will lead to the reduced power of detection and classification. For instance, environmental virologists typically find in many cases of partial RdRp domain (let’s say that contains only the motifs C,D, and E) is partially encoded by the end of short contigs representing incomplete RNA viral genomes; and hence making the algorithm fail at finding it. The same case example could be used to point out how focusing only on the A-B-C palmprint core of the RdRp domain could lead to a faulty classification of an incomplete RNA viral genome. Hence, I encourage the authors to admit such limitations when discussing the scope of the tools they are providing.

Experimental design

The methods used by the authors seem to be rigorous. A minor flaw that I could spot for their reference palmprint datasets for validation and training (L141) is that they used the RdRp in Wolf et al. (2018), but obviously additional novel RNA viral lineages (e.g., leishbunyavirids) have been reported since then, which might be of interest to increase the power of detection. During the identification of viral signal, and to minimise the incorporation of false-positive viral RdRp sequences, the authors even took care of removing the RdRp sequences that were wrongly annotated as viral RdRps in those “decoy” datasets. For benchmarking the classification methods, they compared their protein clusters, generated by dereplication with diverse thresholds, with the ICTV-established taxonomy. Finally, the authors showed the application of their algorithms with the specific case of clinical metagenomics, the discovery of relatives to the Rubella virus.

Validity of the findings

No comment

Additional comments

Minor edits:
- I think that the term “ribovirus”, used throughout the entire manuscript, is wrongly used. Riboviruses are defined typically as “RNA viruses”. Currently the ICTV recognises that RNA virus genomes can code different proteins. The realm Riboviria (“RNA viruses”) contains the kingdom Orthornavirae (RNA viruses replicating via RdRp) and Pararnavirae (viruses replicating via reverse transcriptase). It is clear that the analytics here focused on RdRp, so it targets RNA viruses of the kingdom Orthornavirae, and does not target pararnavirans such as retroviruses; but the term “ribovirus” does not discriminate between orthornavirans and pararnavirans. Please correct this nomenclature issue throughout the whole text.
- similar confusion arises when using the terms “polymerase” or “RNA polymerase”. A “polymerase” could be many types of viral polymerases. Saying “RNA polymerase” is still unspecific since there are also DNA-dependent RNA polymerases”. Please use the names of these replicative enzymes appropriately.
- Please incorporate a space between values and units when writing the lengths of sequences. This mistake is repeated across the text and figures.
- L36-37: no need to write “Fungi” italicised as a formal kingdom, “fungi” can be written non-italicised as a common word. Similarly, the words “Archaea” and “Bacteria” can be written without italicising as “archaea” and “bacteria”; but if italicising as taxa, please specify that they are domains.
- L53: a gene cannot be embedded in an ORF because all genes are ORFs. Rather say that the RdRp domain can be encoded along with other functional traits within multi-domain proteins and polyproteins, encoded by a single ORF, in RNA viral genomes of many lineages.
- To use the acronym “RdRP”, one should write “RNA-dependent RNA Polymerase” instead of “RNA-dependent RNA polymerase”. Why using RdRP instead of RdRp?
- Fig.1 (and in other places), please write viral taxa names italicised every time.
- Fig. 2: although you are describing it in the caption, please write the FULL names of viral phyla, italicised.
- Fig.4: homogenise the number of digits. in the Y-axis of the bottom plot.


BIBLIOGRAPHY
Edgar, Robert C., Jeff Taylor, Victor Lin, Tomer Altman, Pierre Barbera, Dmitry Meleshko, Dan Lohr, et al. 2022. “Petabase-Scale Sequence Alignment Catalyses Viral Discovery.” Nature 602 (7895): 142–47.
Gustavsen, Julia A., Danielle M. Winget, Xi Tian, and Curtis A. Suttle. 2014. “High Temporal and Spatial Diversity in Marine RNA Viruses Implies That They Have an Important Role in Mortality and Structuring Plankton Communities.” Frontiers in Microbiology 5 (DEC). https://doi.org/10.3389/fmicb.2014.00703.
Lauber, Chris, and Alexander E. Gorbalenya. 2012. “Toward Genetics-Based Virus Taxonomy: Comparative Analysis of a Genetics-Based Classification and the Taxonomy of Picornaviruses.” Journal of Virology 86 (7): 3905–15.
Obbard, Darren J., Mang Shi, Katherine E. Roberts, Ben Longdon, and Alice B. Dennis. 2020. “A New Lineage of Segmented RNA Viruses Infecting Animals.” Virus Evolution 6 (1). https://doi.org/10.1093/ve/vez061.
Shi, Mang, Xian Dan Lin, Jun Hua Tian, Liang Jun Chen, Xiao Chen, Ci Xiu Li, Xin Cheng Qin, et al. 2016. “Redefining the Invertebrate RNA Virosphere.” Nature 540 (7634): 539–43.
Simmonds, Peter. 2015. “Methods for Virus Classification and the Challenge of Incorporating Metagenomic Sequence Data.” The Journal of General Virology 96 (Pt 6): 1193–1206.
Starr, Evan P., Erin E. Nuccio, Jennifer Pett-Ridge, Jillian F. Banfield, and Mary K. Firestone. 2019. “Metatranscriptomic Reconstruction Reveals RNA Viruses with the Potential to Shape Carbon Cycling in Soil.” Proceedings of the National Academy of Sciences of the United States of America 116 (51): 25900–908.
Urayama, Syun-Ichi, Yoshihiro Takaki, Shinro Nishi, Yukari Yoshida-Takashima, Shigeru Deguchi, Ken Takai, and Takuro Nunoura. 2018. “Unveiling the RNA Virosphere Associated with Marine Microorganisms.” Molecular Ecology Resources, no. July: 1–12.
Wolf, Yuri I., Darius Kazlauskas, Jaime Iranzo, Adriana Lucía-Sanz, Jens H. Kuhn, Mart Krupovic, Valerian V. Dolja, and Eugene V. Koonin. 2018. “Origins and Evolution of the Global RNA Virome.” mBio 9 (6): 1–31.
Wolf, Yuri I., Sukrit Silas, Yongjie Wang, Shuang Wu, Michael Bocek, Darius Kazlauskas, Mart Krupovic, Andrew Fire, Valerian V. Dolja, and Eugene V. Koonin. 2020. “Doubling of the Known Set of RNA Viruses by Metagenomic Analysis of an Aquatic Virome.” Nature Microbiology 5 (10): 1262–70.
Zayed, Ahmed A., James M. Wainaina, Guillermo Dominguez-Huerta, Eric Pelletier, Jiarong Guo, Mohamed Mohssen, Funing Tian, et al. 2022. “Cryptic and Abundant Marine Viruses at the Evolutionary Origins of Earth’s RNA Virome.” Science 376 (6589): 156–62.

---

## Round 0.2 · accepted · Accept

Thank you for your patience. The manuscript is much improved and I believe it is ready for publication. Your work will be of great interest especially to researchers in the fields of virology and environmental biology. It may help in the discovery of new riboviruses and will hopefully have an impact on metagenomic studies.

All the best

Reviewer 1 ·

Basic reporting

I consider now the manuscript acceptable for publication

Experimental design

I consider now the manuscript acceptable for publication

Validity of the findings

No further comments. I refer to my previous review on the manuscript

Additional comments

No further comments

·

Basic reporting

I think that this current version of the manuscript of Babaian & Edgar accomplishes a substantial improvement.

I appreciate that the authors have answered satisfactorily most of my questions (in addition to having corrected certain of my misunderstandings in their work) and followed most of my recommendations regarding edits and literature.

Only a few of my suggestions given for the sake of improving the clarity for readers have not been incorporated, and I do not think these kinds of stylistic issues are actually relevant for meeting the journal standards or the content itself of this work.

Experimental design

The experiment design of the work meets the standards of the journal.

Validity of the findings

The approach of this work represents a substantial advancement in the field of environmental RNA virology and clinical metagenomics.